# Histone H3 Lysine 4 and 27 Trimethylation Landscape of Human Alzheimer’s Disease

**DOI:** 10.3390/cells11040734

**Published:** 2022-02-19

**Authors:** Giuseppe Persico, Francesca Casciaro, Stefano Amatori, Martina Rusin, Francesco Cantatore, Amalia Perna, Lavinia Alberi Auber, Mirco Fanelli, Marco Giorgio

**Affiliations:** 1Department of Experimental Oncology, IRCCS—European Institute of Oncology, Via Adamello 16, 20139 Milano, Italy; giuseppe.persico@ieo.it (G.P.); martina.rusin@ieo.it (M.R.); 2Department of Biomedical Sciences, University of Padua, Via Ugo Bassi 58/B, 35131 Padova, Italy; francesca.casciaro@unipd.it; 3Molecular Pathology Laboratory “PaoLa”, Department of Biomolecular Sciences, University of Urbino Carlo Bo, Via Arco d’Augusto 2, 61032 Fano (PU), Italy; stefano.amatori@uniurb.it (S.A.); f.cantatore@campus.uniurb.it (F.C.); 4Department of Pathology, School of Medicine, Stanford University, 300 Pasteur Drive, Stanford, CA 94305, USA; amaliap@stanford.edu; 5Swiss Integrative Center of Human Health, Pass. du Cardinal 13, 1700 Fribourg, Switzerland; lavinia.alberi@unifr.ch; 6Department of Medicine, University of Fribourg, Chem. du Musée, 1700 Fribourg, Switzerland

**Keywords:** histone modifications, chromatin, neuronal functions, Alzheimer’s disease

## Abstract

Background: Epigenetic remodeling is emerging as a critical process for both the onset and progression of Alzheimer’s disease (AD), the most common form of neurodegenerative dementia. However, it is not clear to what extent the distribution of histone modifications is involved in AD. Methods: To investigate histone H3 modifications in AD, we compared the genome-wide distributions of H3K4me3 and H3K27me3 in entorhinal cortices from severe sporadic AD patients and from age-matched healthy individuals of both sexes. Results: AD samples were characterized by typical average levels and distributions of the H3K4me3 and H3K27me3 signals. However, AD patients showed a lower H3K4me3 and higher H3K27me3 signal, particularly in males. Interestingly, the genomic sites found differentially trimethylated at the H3K4 between healthy and AD samples involve promoter regions of genes belonging to AD-related pathways such as glutamate receptor signaling. Conclusions: The signatures of H3K4me3 and H3K27me3 identified in AD patients validate the role of epigenetic chromatin remodeling in neurodegenerative disease and shed light on the genomic adaptive mechanisms involved in AD.

## 1. Introduction

Alzheimer’s disease (AD) is the most common form of neurodegenerative dementia and is characterized by extensive synaptic and neuronal loss in areas of the brain essential for cognitive and memory functions, such as the cerebral cortex and hippocampus. Because of this damage, AD patients suffer from memory, cognition and behavioral impairment. The majority of AD cases occur sporadically in adults, while less than 5% of AD cases are familial [1]. AD progresses for approximately 8–10 years after the first diagnosis. Nevertheless, symptoms of cognitive decline assessed by MMSE (Mini-Mental State Examination) occur decades after early biochemical events of neuronal injury set in, calling for earlier targetable biometrics reflecting synaptic dysfunction.

At the molecular level, AD is characterized by extracellular deposition of the amyloid β (Aβ) peptide, a product of amyloid precursor protein (APP) processing, and by intraneuronal neurofibrillary tangles of hyperphosphorylated tau protein. In physiological conditions, APP is cleaved by α-secretase and consequently by the γ-secretases (PSEN); on the contrary, in the amyloidogenic pathway, the β-secretase enzyme cleaves APP to produce a soluble fragment, which is then cleaved by γ-secretase to release the Aβ peptide [2].

During the progression of AD, abnormal tau and Aβ proteins follow distinct sequences of deposition and have been used to define stages/phases (Braak staging for NFT and Thal phases for Aβ) of the disease, measured by postmortem examination of the brain [3].

These patterns of injury are thought to be a consequence of the selective vulnerability to neurofibrillary degeneration exhibited by neuronal subpopulations [4]. In particular, glutamatergic excitatory neurons are highly vulnerable to degeneration, possibly due to their intrinsic susceptibility to glutamate excitotoxicity [5] associated with early synaptic dysfunctions [6]. Molecular mechanisms that trigger excitotoxicity involve alterations in glutamate and calcium metabolism, dysfunctions of glutamate transporters and malfunction of glutamate receptors [7].

In recent years, multiple studies have shown that epigenetic regulation is also involved in the progression of AD. Concerning methylation of DNA, Wang et al. showed a noteworthy age-specific epigenetic drift, with AD patients having a larger epigenetic distance in brain tissue compared to matched controls. This “epigenetic distance” increased with age, suggesting a role of epigenetic mechanisms in the development of late-onset AD [8]. On the other hand, it is less clear if alterations in histone modifications (HMs) are involved in disease onset and progression. Interestingly, age-related changes of H3K27ac and H3K27me3 were detected, suggesting that the global levels of HMs change with age, but it is not affected by amyloid plaque deposition [9].

At the same time, a larger study on the dorsolateral prefrontal cortex of 669 subjects showed genome-wide reorganization of H3K9ac, demonstrating that tau is sufficient to cause such chromatin rearrangements prior to tangle formation [10].

An integrated multi-omics approach, associating transcriptomic, proteomic and epigenomic analyses of postmortem human brains, revealed the upregulation of transcription and chromatin-related genes, including the histone acetyltransferases involved in H3K27 and H3K9 acetylation, suggesting an involvement of the epigenome reconfiguration in AD [11].

Here, we describe the use of chromatin immunoprecipitation (ChIP) coupled with massive parallel sequencing (ChIP-seq) to explore the landscape of two well-characterized transcription-associated histone modifications in the cortex of sporadic Alzheimer’s patients and age-matched healthy individuals. In particular, we investigated both H3K4me3 (which is associated with gene transcription) and H3K27me3 (known to be correlated with transcriptional repression) [12].

## 2. Materials and Methods

### 2.1. Samples

Frozen human entorhinal cortex (*n* = 15) tissues were generously provided by the Medical Research Council Brain Bank for Dementia Research, Oxford (UK), and have previously been described by Bathini et al. [13] and Perna et al. [14]. The use of human tissues has been approved by the Ethical Commission of the Brain Bank for Dementia UK (OBB443 registered 1 May 2017 and OB344 registered 1 February 2017). To assess the mental status of patients, including short and long-term memory, language and communication skills, an MMSE test was performed. It included 30 questions designed to classify the level of mental impairment as mild (score between 21 and 24), moderate (score between 10 and 20) or severe (score less than 10), while patients with a score greater than 25 were classified as normal.

### 2.2. ChIP-Seq

Starting from 10 mg of tissue, samples were homogenized with a Dounce homogenizer in 0.5 mL of fixation solution (5 mM HEPES pH 7.5, 10 mM NaCl, 0.1 mM Na2EDTA, 0.05 mM EGTA, 1% formaldehyde) and fixed for 10 min at 37 °C. After centrifugation, samples were resuspended in 0.5 mL of lysis buffer (10 mM Tris-HCl pH 7.4, 0.15 M NaCl, 3 mM CaCl_2_, 2 mM MgCl_2_, 0.5% Tween20, 1 mM PMSF and 10 μg/mL RNase A—Roche, Mannheim, Germany) and incubated 30 min at room temperature on a rotating platform. When not specified, all centrifugations were performed at 17,860× *g* for 3 min at +4 °C. After resuspension in 0.3 mL of extraction buffer (10 mM Tris-HCl pH 7.4, 0.15 M NaCl, 3 mM CaCl_2_, 2 mM MgCl_2_, 0.1% SDS), samples were sonicated with three pulses of 30 s each, interrupted by 60 s pauses, in a thermoblock refrigerated at −20 °C, with an amplitude of 40% using the EpiShear sonicator (Active Motif, Carlsbad, CA, USA). After being cleared by centrifugation (9500× *g* for 5 min at room temperature), supernatants containing chromatin were saved, and an aliquot of 30 μL was purified using the PCR Purification Kit (QIAGEN, Hilden, Germany) and used for DNA amount estimation by Qubit (Invitrogen, Eugene, OR, USA) using the dsDNA HS Assay Kit (Invitrogen, Eugene, OR, USA) and to control the chromatin size by agarose gel electrophoresis. The chromatin immunoselection was conducted as previously described [15] using anti-H3K4me3 (#39159, Lot. 25817005; Active Motif, Carlsbad, CA, USA) or anti-H3K27me3 (#07-449, Lot. 3091919; Millipore, Temecula, CA, USA) antibodies. Immunoselected DNA, once purified and quantified, was preliminarily used to test the immunoprecipitation specificity by q-PCR and then processed for libraries preparation and, finally, sequenced in 51 bp single-read mode on a HiSeq 2000 sequencer (Illumina Inc., San Diego, CA, USA).

### 2.3. Computational Pipeline

All fastq files were aligned to the Homo sapiens reference genome (assembly NCBI37/hg19) using “bwa” (v0.7.17), a software package for mapping low-divergent sequences against a large reference genome [16]. Using the sample function of bwa, pair-end files were created, and the output (SAM format) files were converted to binary (BAM) format, sorted and indexed using samtools [17]. Unmapped reads, reads with a mapping quality (MAPQ) value smaller than 1, duplicate reads and those that mapped outside of chr 1–22 and ChrX were removed using Samtools. Samtools was also used to additionally remove non-uniquely mapped reads and kept only those marked as properly paired. Resulting uniquely mapped and properly paired reads (stored in a standard BAM format) were used for principal component analysis (PCA) and peak detection performed using MACS2 or Epic2 for H3K4me3 and H3K27me3, respectively [18,19]. Differential binding analysis was conducted with the Diffbind R package (v3.2.6) [20]. Stringency in the analysis was obtained by creating a consensus dataset for each condition, including peaks that were present in at least three samples of the considered group for both HMs. The ChIPseeker R package (v1.28.3) was applied to annotate peak files and DB sites using the curated RefSeq set [21]. Pathway analysis was conducted using the ingenuity pathway analysis (IPA) software from QIAGEN. Pathways with an absolute z-score of >2 were considered significant. For some analyses, BAM files were merged based on group and HM investigated using BAMtools and indexed using SAMtools. Merged bam files were then used to generate a BPM (bins per million) normalized bigwig file (a file format for display of dense, continuous data in a genome browser track) using deepTools bamCoverage (v3.5.1) (ChrX was ignored for normalization). The signal around the TSSs (±5 kB) and along gene bodies was calculated for 77,614 transcripts. The signal, calculated using deepTools computeMatrix, was reported as a mean signal in bins of 10 bp. Missing data were treated as zero. The output was then plotted using plotHeatmap and plotProfile (deepTools) [22].

## 3. Results

### 3.1. Entorhinal Cortex Dataset for Genome-Wide Profile of H3k4me3 and H3k27me3

H3K4me3 and H3K27me3 ChIP-seq data were generated using postmortem entorhinal cortex tissues collected from 15 elderly individuals, as previously described by Bathini et al. [13] and Perna et al. [14] (mean age = 76.2, s.d. = 7.75, range = 61–86, mean postmortem delay = ~41 h, s.d = ~21 h), comprising both AD cases (*n* = 6, mean Braak stage = 5.66, s.d. = 0.52) and age-matched controls (*n* = 9, mean Braak stage = 0.77, s.d. = 0.44; Appendix A). After applying several quality control steps and filters on raw reads, we obtained a mean of 33,505,549 (s.d. = 11,725,251) and 28,734,889 (s.d. = 12,989,317) DNA fragments in pair-end per samples for the H3K4me3 and H3K27me3 profiles, respectively (Appendix A).

### 3.2. Severe Alzheimer’s Disease Has an Impact on H3k4me3 and H3k27me3 Landscape

To estimate the heterogeneity of our datasets, a principal component analysis (PCA) was conducted. To this end, the human reference genome (hg19) was divided into consecutive, non-overlapping bins of 10 kb, and the normalized HM signal was measured for each bin (*n* = 313,763) in each sample. Principal components (PC) 1 and 2 are plotted in Figure 1 for each HM investigated.

The H3K4me3 dataset showed 39% and 15% variability (Figure 1a), while the H3K27me3 dataset was characterized by 29% and 28% of the variability (Figure 1b).

For both HMs, samples belonging to different conditions did not cluster in a specific manner, even if we noticed that in both HM plots, the variability explained by PC1 was driven by the disease while, only in H3K27me3, the second component (PC2) was driven by sex, with males clustering on the upper side of the plot, while females gathered at the bottom. To remove the sex bias and enhance a disease-related trend, we performed PCA using HM signals found only on autosomal chromosomes (number of bins = 298,235) and plotted our results. As expected, the variability of H3K4me3 (Figure 1c) was not affected, enforcing the concept that this HM had a similar profile between males and females. On the contrary, H3K27me3 (Figure 1d)—known to be largely involved in X chromosome inactivation—was strongly affected [23]. Therefore, the PCA relative to all autosomal chromosomes showed a disease-related trend for H3K27me3 similar to the one previously observed for H3K4me3, demonstrating that severe AD is characterized by a distinct epigenetic landscape. Additionally, four PCAs have been produced (Figure 1g,h) based on the sex and HM investigated. Female samples showed a clear distribution based on disease for both HMs investigated (Figure 1g,h); meanwhile, in the male comparison, the control group was not distinguishable from the severe AD group (Figure 1e,f).

### 3.3. Severe AD Patients Show a Decreased H3k4me3 and a Concomitant Increase of H3k27me3 Signal in Gene Loci

Since the two histone modifications investigated played a critical role in transcription control in an opposite manner (H3K4me3 is associated with active promoters whereas H3K27me3 represses transcription), we investigated the signal (normalized by library size and by the investigated region length) across 77,614 transcripts, taking into account a region of 5 kb before the TSS (transcription start site), all gene body (scaling each gene to a common length) and 5kb after the TES (transcription end site). To do this, samples were merged based on groups and HM, and results are shown in Figure 2. Looking at the HM signals at the gene body level, the severe AD group shows an overall decrease in the level of H3K4me3 signal on TSSs (Figure 2a) for the investigated transcripts compared to the control group, while an opposite effect could be found for the H3K27me3 signal (Figure 2b), where the severe AD group is characterized by a higher signal compared to the control group. However, since H3K27me3 is known to be affected by sex (as previously shown in Figure 1b), all the analyses relative to H3K27me3 were split based on sex. As shown in Figure 2c, only severe AD males were characterized by a higher H3K27me3 signal compared to the control group (left panel), while in females, the signal was similar between the two groups (right panel). Overall, this analysis indicates that both HMs are affected by the disease, even if it is in the opposite manner. In particular, severe AD subjects showed a lower intensity signal of H3K4me3 with a concomitant increase of the H3K27me3 signal.

### 3.4. Severe Alzheimer’s Disease Is Characterized by a More Spread in H3k4me3 Signal

As shown in Figure 2a,b, differences between groups could be observed for both HMs investigated; however, as reported in Appendix A, no differences were present in terms of the number of analyzed reads. Starting from this observation, we decided to investigate how the H3K4me3 and H3K27me3 signals are localized across the genome. To this end, using merged files (based on HMs and group) and the PlotFingerprint tool, we retrieved the signal for both HMs across all bins (size of 10 kb). Bins were then ranked on the x-axis based on their signal intensity, and results are plotted in Figure 3a,b. The control group (blue line) was characterized by a more localized signal for H3K4me3, suggesting that in the severe AD group, the H3K4me3 signal was more broadly distributed, while no differences were detected in the H3K27me3 profile. Looking at the H3K4me3 results, it is possible to see that in the control group, the last 10% of bins (i.e., from 90% of genome—x-axis) contained ~45% of reads, while in severe AD, around 30% of reads were contained into the last 10% of the genome. Additionally, a sex-based analysis for the H3K27me3 profile was also performed, but no differences between sexes were detected (data not shown). Our results reveal that in severe AD, the H3K4me3 is characterized by a greater spread in signal than in control.

### 3.5. Severe AD Is Associated with a Reduction of H3k4me3 Signal on Promoter of Genes Involved in Several AD-Related Pathways

Differential binding analysis among groups was carried out in order to identify H3K4me3 genomic loci that could be potentially affected by AD. To this end, we performed peak calling on all filtered BAM files and identified several regions that were then used to create a consensus peak list for both groups to increase the stringency of the differential analysis. We, therefore, investigated differences among groups across 54,800 genomic regions. Trimmed-Mean of M-values (TMM) normalized reads count of all investigated regions were used to compute PCA (Appendix A) and showed that variability of H3K4me3 profiles was unaffected by our genomic loci selection. DiffBind allowed the identification of 937 regions as differential enriched (DE) sites (FDR ≤ 0.1, log2Fc ± 0.5, Conc. ≥ 6—Appendix A) plotted in Figure 4a, and around 48% of them were annotated as promoters (Figure 4b) by ChIPseeker R package. To describe the functional link among these identified sites, we performed an ingenuity pathway analysis (IPA) using only sites that have been previously annotated as promoters. All canonical pathways with *p*-values ≤ 0.05 and z-score > |2| are reported (Figure 4c). Our analysis shows a reduction of the H3K4me3 signal on the promoter of genes whose products are involved in several AD-related pathways. Among these, “CREB Signaling in Neurons” and “Glutamate Receptor Signaling” are the most significant pathways affected by disease, with most of the genes belonging to these terms encoding for proteins localized in the postsynaptic membrane (G-protein coupled receptor or glutamate ionotropic receptor). These findings support the fact that AD is characterized by chromatin rearrangement on the promoter of genes encoding proteins involved in signal transduction. Moreover, IPA allowed us to identify several conditions and functions that have been linked to AD disease. In particular, in severe AD, our analysis showed modulation of the H3K4me3 signal on promoter genes encoding proteins involved in seizure disorder (z-score 3.483), but modulation of the signal was also found on the promoter of genes involved in the transport of molecules, synaptic transmission, learning and cognition (Figure 4d).

### 3.6. AD Affects the H3K27me3 Female Landscape

Differential binding analysis was also performed for H3K27me3 profiles. Since this modification is characterized by a widespread enrichment, peak detection was performed using the epic2 tool. A consensus dataset of peaks was built in order to increase the stringency of the analysis: we considered all peaks present in at least three samples and investigated H3K27me3 profiles for 37,076 genomic regions. Using TMM normalized read counts of all investigated regions, we conducted a PCA (Appendix A) showing that results are comparable to those obtained by using whole genome signals, confirming that biases were not introduced.

However, as already shown in Figure 1b, elevated variability among samples belonging to the same group forced us to split the dataset into two groups: males and females. Moreover, the sample NP13/074 was removed from the female control group due to its distance from the group it belongs to increase the chance to find statistically significant differences (Appendix A). Then, a new consensus peak set was created, taking into account all peaks shared by at least two samples, and a PCA was performed using TMM normalized reads count. PCA results shown in Figure 5a display a clear distinction between the two groups when looking at the first component, which explains 60% of the variability of our dataset. Moving on to the differential analysis, we investigated H3K27me3 female profiles on 33,537 regions finding that, among these, 301 regions were identified as DE sites (FDR ≤ 0.1 log2Fc ± 0.5) and annotated using the ChIPseeker R package. Results plotted in Figure 5b revealed that 44% of these DE sites are located in distal intergenic regions. However, due to the few numbers of DE sites annotated as promoters, we were not able to obtain any significant result using IPA. The male H3K27me3 group was instead analyzed looking at 43,397 genomic regions (consensus peak = 2). Even if the severe AD group is characterized by high variability between subjects (Figure 5c), no samples were eliminated. DiffBind analysis identified only 20 differential enriched sites further annotated using ChIPseeker. As for the H3K27me3 DE sites in females, IPA analysis was not able to identify any functional pathways.

## 4. Discussion

In the present study, we compared the distribution of the trimethylation at K4 and K27 of histone H3 observed in postmortem human biopsies of entorhinal cortex from nine healthy and six AD age-matched patients diagnosed with severe AD. Overall, the results obtained indicate that AD associates with a significant genome-wide redistribution of both H3K4me3 and H3K27me3, allowing us to separate the AD samples from the healthy controls upon principal component analysis. Although the signals of H3K27me3 differ, particularly between sexes, due to the different extent of marking in the extra X chromosome in females [24,25], the H3K27me3 distribution in autosomes was affected by the disease regardless of the limitation of the sample sizes we investigated.

Epigenetic signals, particularly DNA methylation, have been suggested to have a causal role in AD [26,27], although a genome-wide analysis of H3K4me3 and H3K27me3 has not yet been investigated. Thus far, the present study provides the first evidence of neuronal chromatin remodeling in AD, with critical alterations of the histone H3K4 methylation and a concomitant but less pronounced modulation of H3K27me3. Previous reports on human AD brain ChIP-seq findings revealed a consistent effect of AD on histone acetylation associated with the expression of genes involved in the progression of the disease. H3K27 acetylation in entorhinal cortex samples from Alzheimer’s disease was found to be redistributed, particularly within loci related to disease-related genes, such as PSEN1 and PSEN2 [28], whereas in the lateral temporal lobe from AD patients, H3K27 and H3K9 acetylation was found to be globally increased, concurrently with a decreased expression of histone acetyl transferases [11] and H4K16 acetylation decrease [29]. Our analysis of H3K4me3 and H3K27me3 within gene bodies revealed a reduction of the K4 and a concomitant increase of the K27 methylation in the AD group, indicating a heterochromatinization expansion of encoding regions of the genomes associated with neurodegeneration [30].

From the PlotFingerprint analysis of the HMs signals, AD sample results, in particular, were represented by a spread in H3K4me3 signal, suggesting that AD brain tissue includes a larger fraction of nuclei whose active chromatin marking associates less with specific gene elements, like promoters. Differences in cell identity between healthy and AD samples resulted from the ingenuity pathway analysis of differentially bound promoter regions.

Noteworthy, the cellular processes identified by the differentially H3K4me3-marked genes in the two groups are all terms related to critical neuronal signaling functions linked to cognitive decline and, among these, genes clustering to CREB signaling and glutamate receptors (Figure 6) emerged. Regarding these genes, the results obtained indicate that the neuronal composition of the AD cortex has reduced trimethylation of the H3K4 around the TSS. This could originate from negative selection processes associated with brain atrophy and lead to the loss of nuclear open chromatin regions or to a plastic adaptive response involving histone methylases/demethylases that negatively imprint the chromatin of glutamate signaling genes in AD cells.

We are aware of important limitations of the present study related to sample size considering the intrinsic genetic and environmental variability of the human population; however, the small sampled population did not prevent us from identifying distinct HM patterns in AD with respect to healthy controls, as well as important sex influences.

In conclusion, the signature of H3K4me3 and H3K27me3 identified in AD patients validates the role of epigenetic chromatin remodeling in neurodegenerative disease, shedding light on mechanisms involved in AD-related neurodegeneration and guiding to the discovery of novel targets for epigenome-based therapeutics.

## Figures and Tables

**Figure 1 cells-11-00734-f001:**
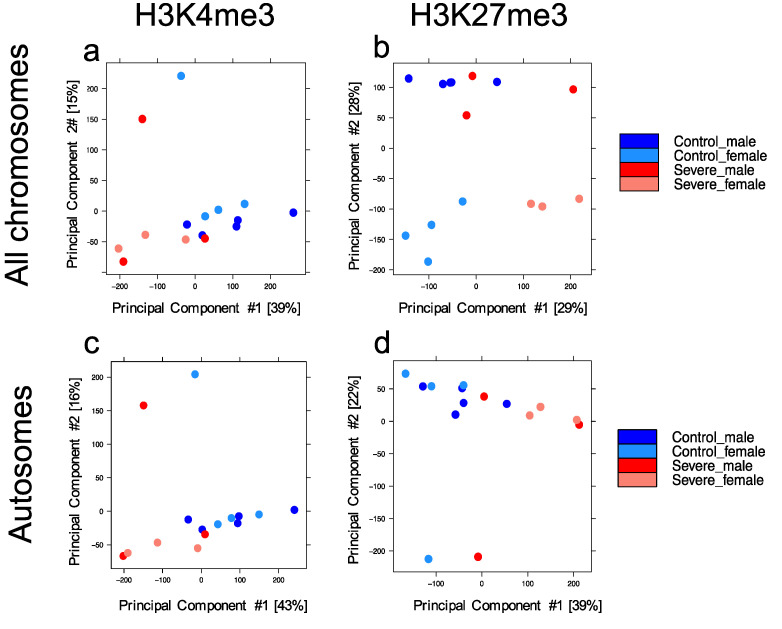
Principal component analysis of genome-wide histone modification signal. Genome-wide normalized histone modification signals were split into consecutive, non-overlapping bins (10 kb). PCA was calculated on the resulting matrix and samples colored by group (control males in blue, control females in light blue, severe AD males in red, severe AD females in pink). (**a**,**b**) Signal across all chromosomes for H3K4me3 and H3K27me3, respectively. (**c**,**d**) Signal across all autosomes for H3K4me3 and H3K27me3, respectively. Signal across all chromosomes in males (**e**,**f**) and females (**g**,**h**) for both histone modifications investigated.

**Figure 2 cells-11-00734-f002:**
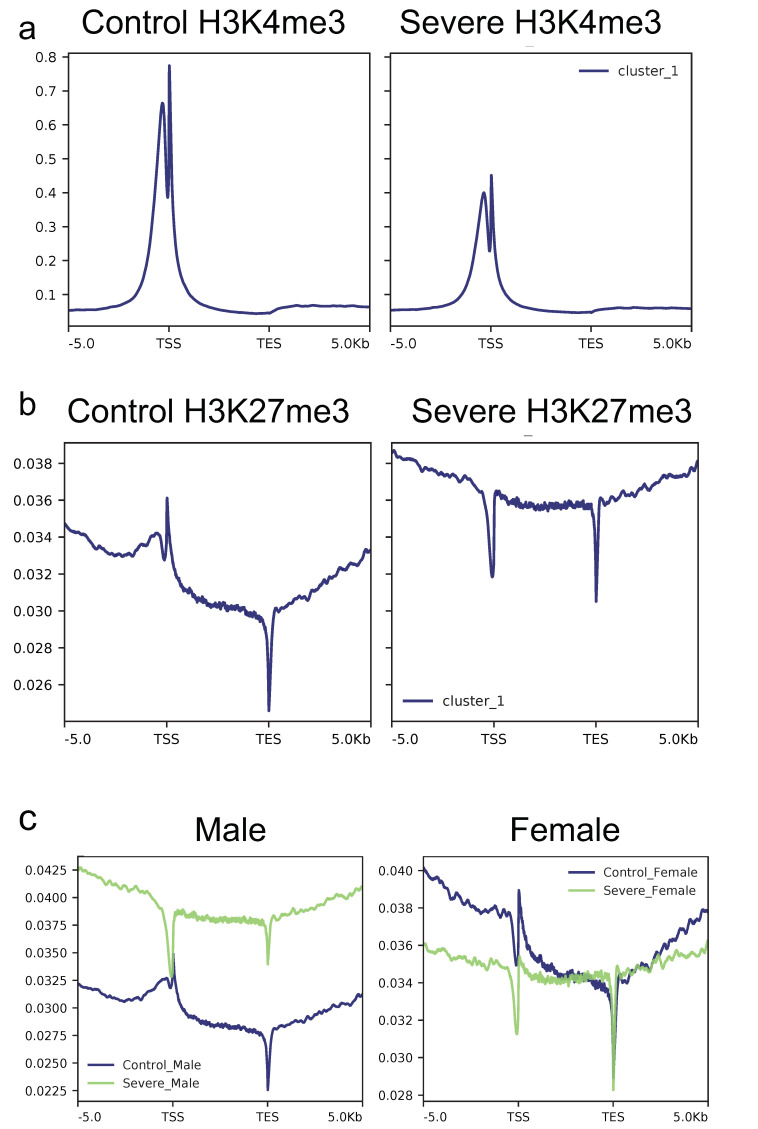
H3K4me3 and H3K27me3 profiles across gene bodies of 77614 transcripts. Samples were merged based on group, and the average normalized profile across the gene body (±5 kb) was calculated. (**a**) The H3K4me3 average profile for the control group shows an overall higher signal than the severe AD group, while an opposite effect is recorded looking at H3K27me3 profiles (**b**). (**c**) H3K27me3 profile was also divided by sex to highlight differences among male (left panel) and female patients (right panel).

**Figure 3 cells-11-00734-f003:**
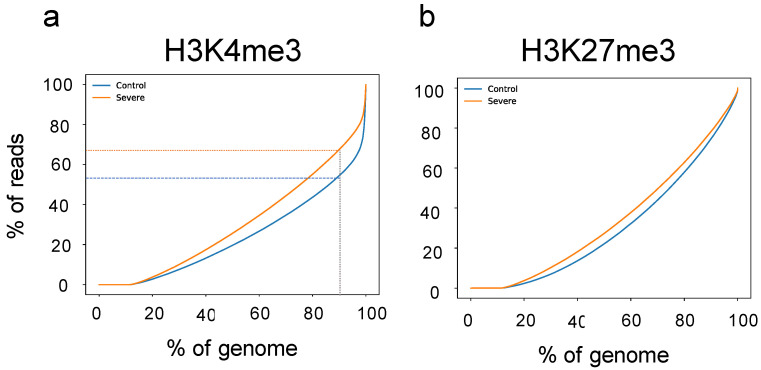
H3K4me3 and H3K27me3 localization at the genome-wide level. Genome-wide normalized histone modification signals split into consecutive non-overlapping bins (10 kb) were retrieved from the group-merged BAM file and ranked on the x-axis. Comparison of H3K4me3 (**a**) and H3K27me3 (**b**) localization signals between control (blue line) and severe AD group (red line). Severe AD group is characterized by a greater spread in the H3K4me3 signal, while no differences between groups are present in H3K27me3 profiles.

**Figure 4 cells-11-00734-f004:**
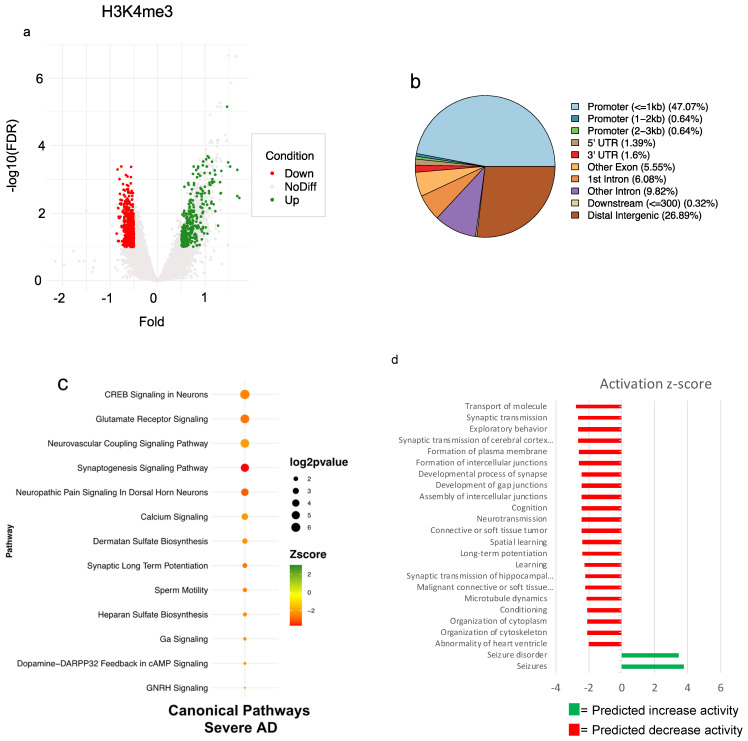
Differential analysis of H3K4me3 profiles identifies the downregulation of several pathways related to AD. (**a**) Volcano plot showing H3K4me3 depleted (in red) and enriched regions (in green) identified by differential analysis. (**b**) Pie-chart showing the annotation of 937 DE site of H3K4me3. (**c**) IPA-identified pathway using all H3K4me3 DE sites previously annotated as promoters. (**d**) IPA function and disease results obtained using all H3K4me3 DE sites previously annotated as promoters. Results show that severe AD patients are characterized by modulation of H4K4me3 signal on promoters of genes encoding for proteins involved in several pathways linked to brain function.

**Figure 5 cells-11-00734-f005:**
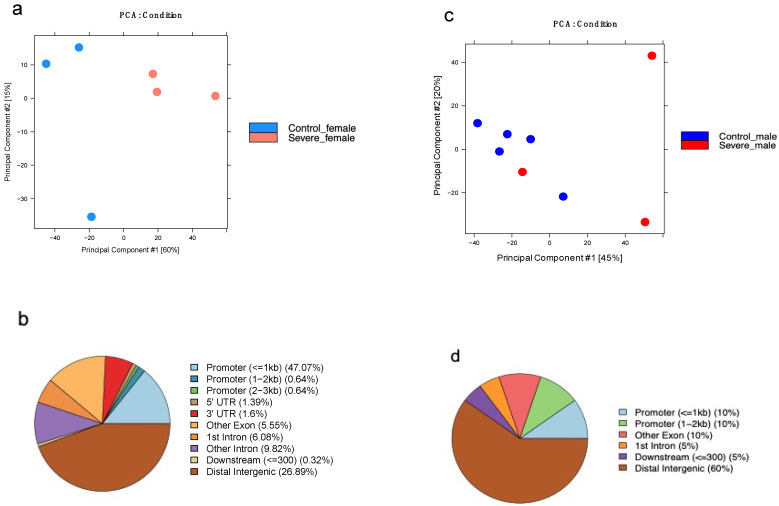
Differential analysis of H3K27me3 profiles based on sex. (**a**) Principal component analysis using TMM normalized read counts of 33,537 regions of female H3K27me3 profiles. Samples are colored based on group (Control in blue, severe AD in red). (**b**) Pie-chart showing the annotation of 301 DE sites of H3K27me3 found after DE analysis for the female group. (**c**) PCA conducted using TMM normalized reads count across 43,397 regions of male H3K27me3. Samples are colored based on group (Control in blues, severe AD in red). (**d**) Pie-chart showing the annotation of 20 DE sites of H3K27me3 found after DE analysis for the male group.

**Figure 6 cells-11-00734-f006:**
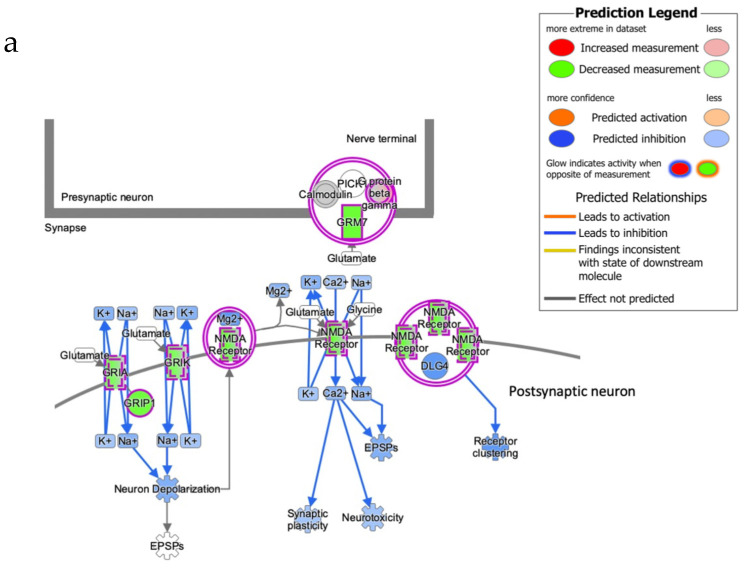
Glutamate receptor signaling and CREB signaling in neurons have generated with IPA. (**a**) Glutamate receptor signaling, (**b**) CREB signaling in a neuron. The intensity of red and green molecule colors is related to the fold change value of each gene included in the pathway. Blue and orange color indicates the predicted inhibition or activation, respectively.

## Data Availability

ChIP-seq data are deposited on the GEO repository and are accessible with number GSE196413 (https://www.ncbi.nlm.nih.gov/geo, accessed on 9 February 2022). All the data that support the figures and other findings are available from the authors upon request.

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
