# Peer review of "Histone H3 Lysine 4 and 27 Trimethylation Landscape of Human Alzheimer’s Disease"

_cells, 2022, doi:10.3390/cells11040734_

Round 1

Reviewer 1 Report

In this paper, Persico et al investigated the genome-wide distribution of 2 histone methylation marks: H3K4m3, H3K27me3, in post-mortem entorhinal cortices of control (n=9) or AD patients (n=6), using chromatin immunoprecipitation coupled to deep sequencing. AD cases investigated were severe cases; The authors performed a series of bio-informatic analyses followed by IPA ingenuity pathway analyses and draw several conclusions on the H3 methylation dysregulations occurring in late stage of AD. If data obtained on H3K4me3 show a quite clear signature, H3K27me3 are more mitigated, especially as low number of patients were investigated and given they observed gender-specific differences (due to the X-inactivation involving this mark). While this is an interesting and original study as so far these marks have not been described at the genome-wide level in AD, some concerns are raised, concerning analyses and data interpretation.

1- The description of the patients should be more detailed, especially as it as only few extra information could be found in Bathini et al 2020 and no detailed information in Perna et al 2021. For example, what was the post-mortem delay for sample collection? The post-mortem delay should be documented in the method section, as histone modifications are highly changing during post-mortem delays (Barrachina M, et al 2012; Jarmasz JS, et al 2019).

Also, regarding the brain samples, from the references cited (Bathini et al 202 and Perna et al 2021), it seems that there were more patients. How were these selected to enter in the study present here?

This is a concern as the number of data presented in this is low (a fact also acknowledged by the authors in the discussion section), but also as the authors wish to draw gender-specific conclusions and thus make sub-groups (which is then n=5 controls versus 3 AD for males and 4 vs 3 for females).

Barrachina M, Moreno J, Villar-Menéndez I, Juvés S, Ferrer I. Histone tail acetylation in brain occurs in an unpredictable fashion after death. Cell Tissue Bank. 2012 Dec;13(4):597-606. doi: 10.1007/s10561-011-9278-9;

Jarmasz JS, Stirton H, Davie JR, Del Bigio MR. DNA methylation and histone post-translational modification stability in post-mortem brain tissue. Clin Epigenetics. 2019 Jan 11;11(1):5. doi: 10.1186/s13148-018-0596-7

2- Figure 2 shows the H3K4me3 and H3K27me3 profiles across gene bodies, as stated, it is the “the average normalized profile”. As it was normalized, how come the start and the end do not overlap (especially in males, H3K27me3), or is this due to a rather global increase of H3K27me3? Can these profiles be shown with a larger window to see if the start and end do overlap (±5Kb, ±10 Kb)? Or can the genomic sites be profiled (independently of whether it is located on a gene or not).

3- Figure 4. 937 regions were identified as differentially enriched in AD vs Controls, the number of depleted and enriched ones should be given and IPA may be led separately (unless all were depleted?). A volcano plot (or similar representation) could be shown.

Figure 4c. Are the up-regulated terms in green? If so, this is not mentioned.

4- If understood correctly, “Glutamate” and “synaptic transmission” are down in H3K4me3 (Fig4b) and “seizures-related terms” are up (Fig4c). How to explain this concerning neuronal hyperexcitability found in AD brains? This should be discussed.

5- From the first mentioned results on H3K27me3 (line 259-260; gene body profiles), more changes were expected in males than in females, so that the authors claim that there is a decrease of H3K27me3 in AD, with strong separation between male and female. However, from the differential analyses, only 20 regions were affected in males and 301 in females (line 475 and 482) and no biological pathways were found to be affected in both genders (line 477-483). How could this be explained? 

Reducing the cutoff of FC would maybe help to define a signature. Of course, increasing the number of samples would be the ideal.

Additionally, are the differential regions depleted or enriched in H3K4me3? Common peaks between the two histone marks studied (so-called “bivalent promoters”) could be investigated. 

Lastly and importantly, the title is then misleading, as H3K27me3 signature in AD does not seem to be clearly defined. 

Minor comments:

6- (line183) “H3K27me3 (Figure 1d) - known to be largely involved in the X chromosome inactivation - is strongly affected”. Reference is missing.

7- Male and female samples should be indicated in the PCAs of Fig1 and FigS2. A difference is observed on males and females so the authors removed sexual chromosomes from the analyses. An independent PCA for males and another one for females could be shown In Figure 1, or 3D PCA could be used as alternative. The removed sample (NP13/074) (line 468) should be identified in the PCA.

8- Lines 344-347. “Our results reveal that in severe AD the H3K4me3 is characterized by a more spread in signal than in control where around 45% of reads are localized into 10% of genome. “. The meaning of this is not clear.

It is difficult to understand these calculations probably referring to figure 3 (and also one is not shown). May be the X and Y axis of graphs from figure 3 should be enlarged and better explained in the Figure legends/Mat & meth. In which conditions were these calculations done (with or without ChrXY?

9- Line 389. What is « Conc= 6 » ?

10- In the discussion section, the authors tend to generalize their finding as for e.g. this sentence (line544-546): “So far, the present study provides the first evidence suggesting that AD is characterized by vast remodeling of the neuronal chromatin by critical alteration of the histone H3 methylation, which is involved in gene expression” but this is not quite true regarding to their data, as they show significant modification only for H3K4me3. Terms like “vast” and generalization on H3 modifications should be avoided.

11- A reference indicating that heterochromatinization expansion requires concomitant reduction of K4 and increase of the K27 methylation should be provided (Line 544-546: “Our analysis of H3K4me3 and H3K27me3 within gene bodies revealed a reduction of the K4 and a concomitant increase of the K27 methylation in the AD group, indicating an heterochromatinization expansion of encoding regions of the genomes by neurodegeneration.”).

12- Discussion: (line 575) “cell loss and infiltration that is typically seen in severe AD, changes the cell composition of the tissue. However, opposite H3K4me3 and H3K27me3 densities suggest that changes are specific to the condition.” - The opposite enrichments do not suggest that the changes are specific to the AD condition, as the contribution of H3K4me3 and H3K27me3 in glia and neurons might be different. Although H3K4me3 changes showed neuronal related terms, no terms were defined in H3K27me3.

13- Figure 6. The figure is not understandable as presented, all red arrows point to a receptor colored (or half-colored?) in green. What does the blue color represent? Does “up-regulated” and “down-regulated” refer to HM-enrichment/depletion or receptor expression?

14- Are the data sets deposited (number?) on GEO-NCBI ?

15- The list of deregulated loci/genes should be given in a table, at least for H3K4me3 which shows significant pathways.

Reviewer 2 Report

In the manuscript Histone H3 lysine 4 and 27 trimethylation signatures associated with human Alzheimer’s disease, the authors investigate epigenetic differences in human entorhinal cortex tissue from healthy and Alzheimer’s disease individuals.  While the investigation is an interest and relevant one, it is my opinion that this manuscript suffers significantly due to issues with clarity, hurting the ability of the reader to follow.  While the key finding, that H3K4me3 and H3K27me3 are different in healthy vs Alzheimer’s disease sample seems convincing enough, the other findings could be significantly improved through the use of statistical analysis and clearer explanation of the relevance of the findings.  Overall, it is this reviewer’s opinion that this manuscript requires some significant revision before it will be ready for publication.  Several specific concerns are outlined below.

Major Concerns

As you recognize in the discussion, this analysis is limited by the smaller sample size.  However, have you performed any type of statistical analysis of these results to show that your key results are significant?

Moderate Concerns

The quality of the images needs to be improved: the font size is too small, the font used is inconsistent within the same figure.  Figure 6, in particular, borders on unreadable.

A sizeable amount of editing for grammar and phrasing is necessary for this manuscript before it is ready for publication, particularly in the Discussion section.

Line 175: Could you please clarify / expand on this statement: “For both HMs, samples belonging to different conditions don't cluster in a specific manner, however a disease-related trend can be observed.”  This seems potentially contradictory.  Is the trend you note solely based on subjective observation, or can you quantify this trend in some way?

Figure 3: Your statement is: “Looking at H3K4me3 results, it is possible to see that 90% of bins (i.e. 90% of genome – x-axis) contains ~55% of reads, while in severe 289 AD around 70% of reads are contained into 90% of the genome.”  If more reads are contained within the same number of bins in the AD group, wouldn’t that suggest less spread in the AD group?  Please explain.  Also, please expand on the biological significance of this finding.

Minor Concerns

Your statement in the Discussion implies that CREB pathways will be outlined in Figure 6 as well, but the figure exclusively outlines glutamate receptor signaling.

Reviewer 3 Report

The manuscript represents an important addition to the AD research field, detailing the alterations in histone lysine methlyation patterns caused by the disease. The text is well-written and easy-to follow, it offers a clear explanation of the observations and their interpretations. The figures are clear and good-quality, although Figure 6. needs updating. Apparently, the red arrows point to genes highlighted in green and there are no genes highlighted in red. It's also advisable to increase the size of the legends, especially in the case of the processes affected. The blue colour scheme is not explained either, that adds to the confusing nature of this figure.

Round 2

Reviewer 1 Report

The authors have properly addressed to my comments.

Reviewer 2 Report

The authors have taken the time to address the concerns of myself and the other reviewers.  While some grammatical editing would still be beneficial, this manuscript has improved and is ready for publication.